# Indicators to Measure Efficiency in Circular Economies

**Jaime Sánchez-Ortiz ***[ID]**, Vanesa Rodríguez-Cornejo, Rosario Del Río-Sánchez**[ID] **and Teresa García-Valderrama**

Faculty of Economics and Business Administration, University of Cádiz, Avda. Duque de Nájera, 8, 11002-Cádiz, Spain; vanesa.rodriguez@uca.es (V.R.-C.); rosario.delrio@uca.es (R.D.R.-S.); teresa.garcia@uca.es (T.G.-V.)

**\*** Correspondence: jaime.sanchez@uca.es; Tel.: +34-956-015-180

**Abstract:** In this paper, a number of indicators are shown to measure economic efficiency in terms of circular economy (CE). The European Union affirms the need for a comprehensive model of indicators relating to CE in order to meet the needs of all participants (individual companies and industry, society, and the nation), to be based on three perspectives: environmental impact, economic benefit, and resource scarcity. Therefore, the objective of this work is to define these indicators and establish models for measuring the efficiency of processes and products of CE (through Data Envelopment Analysis, (DEA)) in its different manifestations. The models will be useful for both organizations and external users in relation to CE in order to facilitate the search for indicators for all users. Following the bibliographic review of official reports and different high impact works, our results demonstrate the ability to obtain information concerning the main indicators of CE and how the efficiency of CE models has been measured through the most frequently used inputs and outputs.

**Keywords:** circular economy; indicators; efficiency; data envelopment analysis

## 1. Introduction

The current production model has become almost unsustainable due mainly to a series of consequences that originate from it, such as high greenhouse gas emissions and increased competition for access to increasingly scarce resources, which increase the risk of supply chains and have generated high volatility in the price of raw materials as well as large quantities of waste.

Society today has become aware of the current situation regarding the planet; hence, several organizations worldwide have successfully carried out research into the application of circular economy (CE) principles [1], in which they have achieved optimal production with minimum consumption of natural resources, while minimizing the emission of greenhouse gases and waste by reusing and recycling that waste.

The European Commission [2] adopted an action plan for circular economies in order to give a new boost to employment, growth, and investment, and to develop a carbon-free, resource-efficient, and competitive economy. The monitoring framework for the circular economy of the European Union (EU) [3] shows that the transition has helped to create jobs. In 2016, the sectors relevant to circular economy employed more than four million workers [4], an increase of 6% compared to 2012. Circularity has also created new business opportunities, led to the emergence of new business models, and developed new markets, both at the national level and outside the EU. In 2018, circular activities such as repair, re-use, or recycling generated an added value of almost 154 billion of euros [5].

The European Union affirms the need for a comprehensive model of indicators relating to CE in order to meet the needs of all participants (individual companies and industry, society, and the nation) and be based on three perspectives: environmental impact, economic benefit, and resource scarcity [6].

Therefore, the objective of this work is to define these indicators and establish models for measuring the efficiency of the processes and products of circular economies in their different manifestations;

models that are useful both for organizations and for external users who are related to it, in order to facilitate the search for indicators for all users who are related to circular economies.

Therefore, in this study, after defining the concept of circular economy, the need to measure efficiency in those organizations that apply CE principles has been raised. This measurement will be carried out through indicators whose definition and establishment are not exempt from difficulty. Due to this, we then analyze the problems in measuring CE efficiency and the limitations in defining indicators that quantify efficiency. Finally, we propose a bibliographical review of the main efficiency models in circular economies, as well as the indicators that have been used. Finally, the most relevant conclusions of the study are presented.

## 2. Circular Economy: Evolution of Concept

The current production model has become almost unsustainable because of high greenhouse gas emissions and increasing competition for access to increasingly scarce resources, which increases the risk for supply chains and has generated high volatility in the price of raw materials [7], as well as the large amounts of waste that it generates. The concept of CE, as noted by Stahel [8], is a spiral loop within which an attempt is made to minimize energy consumption and environmental deterioration without slowing down the growth of organizations. This concept has resurfaced in the face of the need to change the relationship between people and the material world and the awareness that people have of the need to reduce production and increase recycling [9–12].

CE is one of the most relevant issues addressed by public bodies and is of great concern to both businesses and end consumers [13,14]. It is also of great interest to both academics and professionals as it addresses a necessary condition for sustainable development in business [15,16].

This is a concept that has not been clearly defined, and on several occasions has been confused with the concept of green economy [17], which has a focus on environmental, ecological, and sustainable development issues. For example, Lieder and Rashid [12] note that there are various possibilities for defining CE, while Yuan et al. [18] note that no commonly accepted definition of circular economy exists. However, in this section, we aim to clarify the concept of CE, its evolution, and its usefulness for both society and the business world.

The concept of CE has its origin in a double scientific aspect: On the one hand in the field of engineering, centered on research related to industrial ecology [19–21], and on the other hand, in the field of ecological economics, centered on research related to recycling and efficient use of waste [22–26]. In turn, CE is related to other more specific fields such as industrial ecosystems [27], industrial symbiosis [28], clean and efficient production [12,15,29], eco-efficiency [30–32], eco-efficient design [33,34], performance economics [35,36], or reduction of pollutant emissions [37], among others.

One of the most current definitions of CE is that provided by Korhonen et al. [38], which identifies it as a sustainable development initiative aimed at reducing the linearity of production-consumption systems and reducing the flow of materials and energy used in the manufacture of outputs. CE therefore promotes an adequate use of material cycles, together with traditional recycling and the development of cooperation systems between producers, consumers, and other social actors to promote sustainable development.

Hobson [39] has defined CE as an industrial system that regenerates industrial production design by replacing the concept of end-of-life (EOL), moving towards renewable energy use and eliminating the use of products that harm the environment through changes in business models. Singh and Ordoñez [40] add to the previous definition of CE in noting that CE is a strategy that seeks to modify the linear system of consumption to a circular system in order to achieve economic sustainability.

Moreau et al. [41] and Haupt et al. [42] provide a very similar definition of CE, explaining it as a concept that guarantees a production and consumption system with minimum losses of materials and energy through reuse, recycling, and recovery. Naustdalslid [43] and Blomsma and Brennan [44] complete the definition presented above by incorporating the concept of circulation, i.e., that it is a

productive process in which the circulation of materials in the economic activities of production and consumption becomes the key element for understanding the concept of CE.

Kirchherr at al. [45] state that CE is presented as a combination of reducing, reusing, and recycling activities, without taking into account the need for a systemic change of the concept, while finding few definitions that relate the circular economy to sustainability and development [46]. It is noteworthy that the purpose of the existence of CE is economic prosperity, environmental quality, and the impact on social equity and future generations.

Therefore, the main objective of a CE is that the natural resources consumed in production have an unlimited life through the reuse of the waste generated by the production process itself and of the products, once these become waste. This could partly solve the growing demand for increasingly scarce natural resources.

A CE offers effective solutions, as it harmonizes ambitions for economic growth and environmental protection [12]. By closing the cycles of matter, water, and energy, it enables the economy to grow while reducing extractions from the natural environment, thus automatically converting the waste of some into the resources of others and becoming the economy of recovery, reuse, and recreation [47].

This trend is taking on great importance and perhaps this is because society is now aware of the situation the planet is in, making CE a relevant and prevalent concept in both politics and business development [48], as it transforms economic activities from carbon-intensive manufacturing to more sustainable production and consumption. Furthermore, CE is an important concept, not only for researchers and companies, but also for society, and it is necessary to bring together models that can measure the levels of CE in companies through efficiency in order to increase business sustainability.

At the institutional level, the EU has made significant progress in the field of CE. The European Commission has issued several communications with the aim of having the different member countries legislate and implement the necessary policies to facilitate the application of the principles of CE in different sectors of the economy [2,3]. Converting CE into a current project that society in general and companies in particular must apply is important for two main reasons: To preserve the environment and for legal imperatives.

Figure 1 explains the model of Lieder and Raschid [12], which illustrates the commitment and interaction that must exist between governments, society, and the business sector.

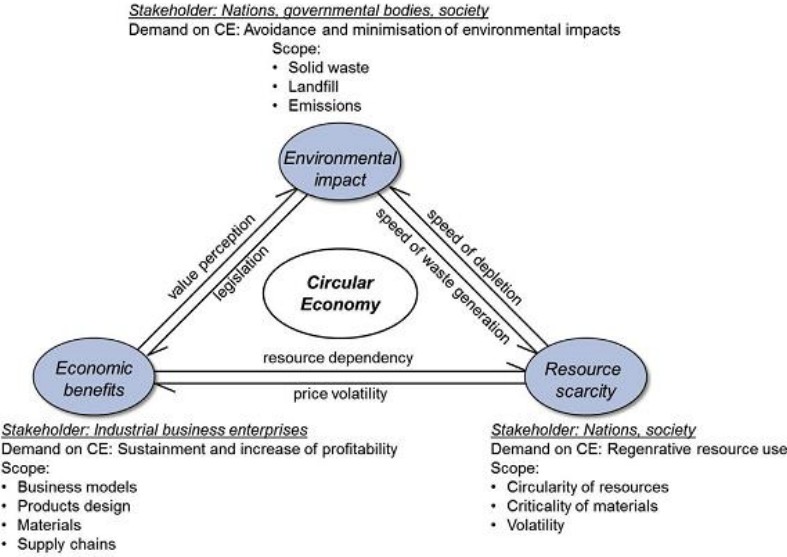

**Figure 1.** Integral approach of a Circular Economy (CE).

We are facing an economic model that must transform society so that it can survive in the future; hence, systems for evaluating the measures taken by organizations concerning circular economy are essential. There is no point in implementing waste recovery measures, for example, if we are not

able to measure the efficiency of these measures and, logically, improve them through the feedback provided by reliable indicators adapted to the new context in which companies have to develop.

For this new management model there must be: necessary indicators that integrate these objectives; management models that help to make decisions focused, not so much on immediate profitability, as on stable and sustained growth; and recovery of waste and products that generate new lines of business or are reincorporated into already developed production processes.

## 3. Problems in Measuring Efficiency in a Circular Economy

The definition of efficiency indicators in CE requires high complexity [49]. According to the European Advisory Council for the Academic Sciences [50], researchers find it difficult to establish indicators that measure levels of CE performance in organizations, i.e., to define indicators that measure the reduction, reuse, and recycling of waste. This is why it is necessary to inform entrepreneurs and researchers on how they can evaluate the impact of CE on the efficiency levels of organizations.

Bocken et al. [51] state that, on many occasions, companies cannot propose solutions to problems derived from CE because of the scarcity of indicators and targets, i.e., because of the lack of knowledge about the alternatives produced by CE and its economic benefits to the business world and society in general, as it is a new scientific branch of study. Haas et al. [30] explain the need to establish a series of reliable indicators as tools to measure and quantify the improvements generated by CE. This statement is supported by The European Commission, which has also recognized this need for circularity indicators through its action plan for the European Union [2], stating that "to assess progress towards a more circular economy and the effectiveness of action at national and EU level, it is important to have a set of reliable indicators".

Therefore, this section shows what dimensions a manager should take into account when defining a CE indicator in order to solve the main problems that various authors have encountered when measuring levels of efficiency in CE, so that without adequate knowledge and indicators, there will always be certain limitations when measuring the impact of CE on efficiency [52]. Linder et al. [53] underline an urgent need to explicitly review the solutions available for measuring circularity, in order to find solutions to its different weaknesses, or to identify some complementarities.

In response to this growing number of diffuse and complex indicators in a CE, in this section we aim to explain, through an exhaustive bibliographic review, the different phases that researchers and entrepreneurs must take into account in order to adequately define this type of indicator: First, we explain how to define an indicator; second, we define the dimensions that must be taken into account in order to define that indicator; and third, we explain the different limitations that researchers have encountered when measuring CE.

Specifically, with respect to CE, indicators must have a purpose directed towards CE practices, so that they really measure the desired impact on efficiency as a springboard for a transition towards the use of new CE practices [54]. Therefore, their different potential uses [53], the level of performance they report, as well as the impact of regulation on this type of activity, should be taken into account as key performance indicators [55]. In response to the complexity of measuring these indicators, the researcher must rely on the interrelationships of the different phases of the company's value chain, providing indicators that include the implementation of these activities [56]. Defining good indicators in CE allows for the quantification of their effects on the organization's efficiency and is a source of relevant information for managers and entrepreneurs in their decision-making [57].

Given the number and diversity of sustainability indicators that have been developed, it was becoming increasingly difficult for managers, when carrying out their decision-making, to grasp the meaning and relevance of that indicator and to quantify the impact of the indicator on their efficiency levels. Measuring elements such as the environmental footprint, the recycling quota, or the environmental effects in waste management involves great complexity for researchers and company managers.

However, Park and Kremer [58], despite the variety of existing resources, have aimed to establish sustainability indicators that facilitate the quantification of the impact of CE on efficiency levels for the mining sector. Those authors extracted sustainability indicators from the existing literature grouped into four relevant categories to clarify their use and facilitate their application in companies: (i) environmental (chemical impact and release), (ii) pollution from emissions and waste, related to end-of-life and use of chemicals, (iii) raw material and facility management indicators, and (iv) energy and water management.

With respect to eco-design, another major dimension of CE, Bovea and Perez-Belis [59] reviewed and ranked eco-design. They used assessment tools to facilitate the integration of the product into the design process. With the intention of providing designers with a guide to select the eco-design tool that best suits a company, these authors established the series of fields needed to define indicators related to eco-design: (i) designing a method for environmental assessment; (ii) defining the product as well as the requirements to be integrated into it in addition to the environmental requirements; (iii) taking into account the product's life cycle (to measure the impact of reuse); (iv) the qualitative and quantitative nature of environmental assessment; and (v) the stages of product manufacturing design.

Another of the relevant dimensions for measuring CE in organizations is the business model. Lewandowski [60] established a series of guidelines for measuring CE in terms of the business model, by analyzing a sample of 20 types of businesses that implemented CE in their organizations. With this work, he managed to establish a business model structure that would provide a canvas for quantifying the effects of CE.

More recently, Urbinati et al. [61] proposed a taxonomy of business models based on CE and its impact on efficiency levels, identifying two dimensions to define indicators: (i) the customer value proposition and interface and (ii) the value network. Lüdeke-Freund et al. [62] conducted a review and analysis of 26 CE businesses, identifying six major dimensions: (i) repair and maintenance; (ii) reuse and redistribution; (iii) refurbishment and remanufacturing; (iv) level of recycling; (v) level of reuse; and (vi) organic raw material business model patterns.

Wisse [63] described an overview of deficiencies in the literature on the assessment, measurement, and implementation of CE in organizations. The main problems he had in his research were the lack of knowledge and practice in the indicator framework, the low level of stakeholder participation in the indicator design process, and the fact that certain indicators represented holistic fields.

Rossi et al. [64] established the main barriers that prevent an adequate implementation of the circular manufacturing process in organizations and proposed a series of ideas to overcome these barriers. Bovea and Perez-Belis [59] stated that most eco-design tools cannot be applied systematically, which increases the level of complexity of CE models and their implementation time. However, they explain that it would be necessary to establish a set of common indicators to measure these levels of efficiency to serve as a guide for defining organization-specific CE indicators.

Park and Kremer [58] assert that companies face problems in measuring the efficiency of CE because they do not know the relevance and potential environmental benefits. However, they state that there is a lack of information regarding the usefulness of the indicators and the theoretical and technical aspects that define them. In addition, these authors explain the difficulty of indicators having a direct effect on business practice.

Another limitation that makes it necessary to establish a set of common indicators to measure efficiency in CE is the collection of data. Potting et al. [49] state that data on CE are a relevant barrier to adequately defining an indicator, because they imply a high search time and a high economic cost for companies. Birat [65], in relation to this limitation, explains the difficulty of collecting data due to the lack of exchange of information between researchers and company managers, resulting from the confidential aspects of the data or concepts that affect CE.

Therefore, subsequently, we aim to resolve one of the limitations raised in this section, which consists of establishing a series of efficiency indicators that make it possible to quantify (in a generic way) the dimensions of CE.

## 4. Choice of Indicators to Measure Efficiency in Circular Economy

There is no point in having CE principles marked within the purposes of an organization if the managers of the organization do not have adequate indicators to measure the results of their decisions in relation to the best management of the company. Only the development of specific CE indicators and integrated management models will allow them to efficiently change the classic management of companies towards the new economic paradigm where, undoubtedly, CE becomes a necessity.

The European Commission [4] highlights the importance of developing a monitoring framework that aims to measure progress towards CE in a way that covers the various dimensions at all stages of the life cycle of resources, products, and services. The monitoring framework should contain a set of indicators grouped into four stages and aspects of circular economy: (1) production and consumption, (2) waste management, (3) secondary raw materials, and (4) competitiveness and innovation. This is broadly in line with the logic and structure of the action plan for CE.

For the elaboration of these indicators, the production process of the CE, i.e., the overall material flow balance, has to be taken into account (Figure 2):

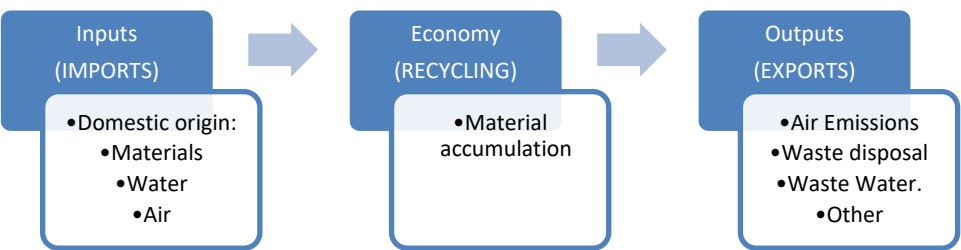

**Figure 2.** Simplified material flow balance scheme (including air and water).

For this new management model, indicators are therefore needed that integrate these objectives. This would be a management model that helps in decision-making, focused not so much on immediate profitability as on stable and sustained growth and recovery of waste and by-products that generate new lines of business or are reincorporated into the already developed production processes.

Thus, each of the participants, depending on their needs, will demand indicators that focus on different aspects. On the other hand, when designing a system of indicators to evaluate CE, it must be developed considering different levels: micro-, meso-, and macro-:

- At the micro-level, each company needs to design a set of specific indicators according to the characteristics of the company, conditions, and existing problems. Indicators are usually used that are based on the 3Rs principle of waste (reduce, reuse, recover) but not on CE in general.
- At the meso-level, the concept on which measurement focuses is that of industrial symbiosis, consisting of the use of common infrastructures and services, i.e., indicators that help to control the performance of plants and industrial parks.
- Finally, at the macro-level, it is a matter of designing indicators to evaluate, monitor, and improve policies.

In order to establish a system of indicators at any of the previous levels, we must follow a specific routine. Valerio, Grazia-Gnoni, and Tornese [66] propose the following steps for the verification of a CE strategy:

1. Observe the system and processes to be analyzed. This can be a single process, multiple processes, or the entire supply chain. For example, a zero waste strategy focuses on managing final resources.
2. Identify the activities that will serve to determine which requirements to measure.
3. Choose the methodology to be applied.

To do this, it is important to analyze the business model of companies involved in CE, because a business model is a conceptual tool that helps to understand how a company works and can be used for analysis, performance assessment, comparison, management, communication, and innovation [67].

Business model innovation is recognized as the key to producing greater social and environmental sustainability in the industrial system [68]. In CE, up to eight business model archetypes are recognized, depending on the type of innovation that has been made: technological, social, or organizational [69]. These eight archetypes are summarized as follows:

1.　Technological innovations:

  - Maximize material and energy efficiency;
  - Create value from waste;
  - Replace current processes with renewable and natural processes.

2.　Social innovations:

  - Give functionality more than property;
  - Take a proactive role;
  - Encourage sufficiency.

3.　Organizational innovations:

  - Change the purpose of the company with society and the environment in mind;
  - Generate scalable solutions.

Each archetype will need specific indicators to measure its performance, input indicators, and output indicators, both at the micro- (company) and meso- (industry) levels and at the macro- (region, country) level. The indicators will be used to see the scope of CE and measure its efficiency.

Below, we show in Table 1 some of the works published in the specialized literature, where dimensions and measurement indicators are shown for the three levels of micro-, meso- and macro-. The selection of the works has been carried out by way of the impact factor of the published journal and according to the indicators included in the documents published by the European Commission. For this, we have searched for papers cited in Journal Citations Reports (JCRs) and Scopus using the keywords "circular economy, DEA". In addition, we have selected the papers with the highest number of citations.

**Table 1.** Indicators used to measure different aspects of the Circular Economy.

| Authors (Years) | Indicator (Dimension to Which It Belongs) | Definition |
|---|---|---|
| Zhijun and Nailing [70]. | Indicators on CE influenced by harmonized economic, social, and ecological relationships, i.e., an index system for assessment must incorporate these three dimensions | Production indicators (land-product ratio). Reuse indicators (water reuse) Resource index (emission of industrial gases or solid waste) |
| Geng et al. [71]. | Performance measures in eco-industrial parks: Economic development indicators (2 indicators); Indicators for the reduction of materials and recycling (7 indicators); Indicators for pollution control (8 indicators); Indicators for park management (21 indicators). | Economic development indicators (2 indicators): industrial value added per capita or growth rate of industry (%) Indicators for the reduction of materials and recycling (7 indicators): solid waste generated by industry (tonnes/yen), solid waste generated (tonnes/yen), or freshwater consumption by industry ($m^3$/yen). Indicators for pollution control (8 indicators): $SO_2$ emitted by industries (kg/yen), ratio of hazardous solid waste disposed of (%), or level of domestic waste used (%). Indicators for park management (4 indicators): quality of the environmental report (yes/no) or degree of satisfaction of the local environmental quality (%) |
| Geng et al. [72]. | Indicators to measure the impact of CE. Indicators at the macro-level (22 indicators). Indicators at the industrial park level (11 indicators). | Macro-level indicators related to external resources used (energy resources, mineral resources), recycling, and water consumed (energy consumed as a function of GDP, water extraction per unit of GDP, water and energy consumption per unit of product in industrial consumption), and pollutant emissions (Total $SO_2$ emitted or industrial wastewater emitted). Indicators at industrial park level (11 indicators): consumption of internal resources (land, energy, water), efficient use of internal resources (energy and water consumed by industrial production, energy and water by unit of product), pollutant emissions (quantity of wastewater and solid rights emitted), and water used in that specific park (proportion of reuse of water used). |
| Su et al. [73]. | Evaluation of CE Indicators proposed by the China National Development and Reform Commission (at the meso-level) (13 indicators). Indicators proposed by the Chinese Ministry of Environmental Protection (at the meso-level) (21 indicators). | Indicators proposed by the China National Development and Reform Commission (13 indicators): Production indicators (production of water, land, energy, etc.), level of resource consumption (energy consumed per unit of production or water consumed per unit of production), integrated resource use (level of resource recycling), and reduction in waste generation (solid waste use and wastewater generation rate). Indicators proposed by the Chinese Ministry of Environmental Protection: Economic development indicators (2 indicators) (Industrial value added per capita, industrial value growth rate) Indicators of raw material use and recycling (7 indicators): energy consumption per industrial value added, industrial water reuse, wastewater generation per unit of industrial value added, indicators of pollution control (8 indicators): $SO_2$ emission per unit of industrial value added, rate of hazardous solid waste disposal, safe treatment of household waste or chemical load of oxygen demand per unit of industrial value added. Waste administration and management indicators (4 indicators): quality of the environmental report, degree of satisfaction of local environmental quality or transparency in the information platform. |

**Table 1.** *Cont.*

| Authors (years) | Indicator (dimension to which it belongs) | Definition |
|---|---|---|
| Zaman and Lehmann [74]. | A tool is defined to measure the performance of waste management systems called the "zero waste index". Through this tool, the aim is to calculate the level of waste of a certain population (measuring the behavior of the environment). | This tool takes into account the following fields: level of environmental awareness and education (impact of zero waste programs), new infrastructure (index of non-polluting technologies), recycling and recovery (level of recycling of a population), sustainable consumption (behavioral change, lifestyles), industry consumption (level of clean production and level of corporate social responsibility), level of legislation and policy (assessment of environmental legislation and government incentives to achieve "zero waste"). |
| Geng et al. [75]. | Indicators based on an Energy analysis. Meso-level analysis (industrial park) (9 indicators). | The indicators (measured in grams/year) deal with the level of renewable inputs (air, rain, solar energy), internal electricity production (coal production and wind production), non-renewable inputs (gravel, clay, groundwater), polluting energy resources (gasoline, diesel, coal), metals (steel, zinc, methanol, etc.), non-metallic polluting products (paper, garbage, plastic), organic food (rice, fruit, vegetables), labor, service, and the outputs obtained (building materials, food products). |
| Park and Chertow [76]. | Indicator of potential reuse in the case of waste generated by coal combustion to quantify the usability of the waste. | The reuse potential indicator expresses the utility of the material by an actual value between 0 and 1. It is equal to 0 when all materials are discarded and 1 when all materials can be reused. It is measured by taking into account the consumption of that product according to its level of reuse. |
| Chen, Liu and Hu, [77]. | Company evaluation indicators:<br>-Economic development (5 indicators)<br>-Social and personal well-being (3 indicators).<br>-Resource consumption (2 indicators)<br>-Recycling of resources (2 indicators)<br>-Environmental quality (3 indicators.<br>-Pollution control (5 indicators) | The indicators used to measure each of the above dimensions are: economic development (Gross domestic product per capita, proportion of tertiary industry (%), weight of high technology in GDP (%)), social welfare (Household disposable income in monetary units, level of urbanised development (%), net annual income of rural households), resource consumption (Energy consumed as a function of GDP, Water consumption per unit of value added in m3/10,000 yuan), resource recycling (rate of industrial water use for irrigation (%) and ratio of industrial solid waste used (%)), environmental quality (rate of green area coverage (%), air quality (%), rate of centralized wastewater disposal (%)) and pollution control (co2 emission as a function of GDP (tons /10,000 john), amount of SO2 emissions (tons) and percentage of harmless treatment for living waste (%)). |
| Golinska et al. [78]. | Assess the level of sustainability of the remanufacturing processes:<br>Economic performance (6 indicators);<br>Environmental performance (4 indicators);<br>Social performance (5 indicators). | The indicators used by these authors are based on economic performance (overall effectiveness of the equipment, remanufacturing process flow, adequacy of remanufacturing process planning, availability of machines and tools, level of service and level of stocks consumed), environmental performance (amount of energy consumed, level of waste generated, material recovery rate, $CO_2$ emissions), and finally social performance (employment, employee training, harmfulness of the manufacturing process, level of comfort at work, and level of innovation). |
| Wen and Meng, [79]. | Indicators based on resource productivity combined with substance flow analysis (SFA) to assess the contribution of industrial symbiosis to the development of CE. | The inputs are the total copper that has been discarded (with a sample of 8 companies) and the amount of copper regenerated by these same companies to understand the waste losses that have existed. |

| Authors (years) | Indicator (dimension to which it belongs) | Definition |
|---|---|---|
| Franklin-Johnson, Figge, and Canning [80]. | Longevity indicator composed of three indicators: initial duration, duration gained by renewal, and duration gained by recycling to evaluate the performance of CE. | To measure the initial duration, the life cycle of the product is taken until it is consumed; for the duration by renovation, the duration of the product is taken into account if modifications are made to it, and finally the duration of recycling is the duration of the product after allowing its use on several occasions. |
| Fundación COTEC para la Innovación [81]. | Policy issues related to progress towards CE from a material perspective (Dimensions): Material input (5 indicators); Eco-design (4 indicators); Production (4 indicators); Consumption (6 indicators), Waste recycling (5 indicators) | The indicators used are material inputs (indicators related to the consumption of raw materials), eco-design (general effectiveness of the equipment, remanufacturing process flow, adequacy of remanufacturing process planning, availability of machines and tools, level of service and level of stocks consumed), production indicators (the production obtained is quantified according to the level of material input production), consumption indicators (energy consumption by industrial value added, reuse of industrial water, generation of wastewater by industrial value added unit), and waste recycling indicators (solid waste generated by industry (tonnes/€), solid waste generated (tonnes/€), or freshwater consumption by the industry (m3/€). |
| Golinska-Dawson et al. [82]. | The remanufacturing process is characterized by a high level of uncertainty regarding the time, quality, and quantity of its products. This study has shown that a simple method for assessing sustainability dedicated to SMEs (Small and medium-sized enterprises) in the remanufacturing sector is missing | To assess the sustainability of SMEs, the authors use indicators related to the polluting effects (calculation of $CO_2$ and $SO_2$ emissions, amount of waste according to the raw materials used, or sustainability performance of SMEs, i.e., discarded factors according to the units produced). |

## 5. Circular Economy Indicators

Once the indicators for measuring efficiency in circular economy have been established, it is relevant to look at the availability of these indicators. As we have analyzed, there is a pressing need to measure the efforts made by companies that practice CE principles, in order to have specific performance measures, assess their efficiency both globally and by processes, and make decisions to improve them economically, environmentally, and socially.

Table 2 provides a series of relevant indicators for measuring circular economy. Through the information provided by COTEC [81] on the level of availability of information of indicators of CE, we have established an evaluation of the availability of the main indicators of CE to provide researchers with useful information so that, if a certain indicator is not highly available, the researchers can seek a subrogate.

**Table 2.** Indicators for measuring aspects of the Circular Economy.

| Possible Indicators | What It Measures | Data Availability |
|---|---|---|
| Direct material consumption or consumption of raw materials. | Are primary material inputs in Europe decreasing? | ++ |
| Proportion of material losses in key material cycles. | Are material losses in Europe being reduced? | + |
| Waste diversion from landfills (EEA indicator WST006, under development). | | ++ |
| Proportion of secondary raw materials in material consumption | Is the proportion of recycled materials in the form of inputs increasing in Europe? | + |
| Proportion of ecologically certified materials in material use. | Are the materials used in Europe obtained in a sustainable way? | + |
| Durability or life cycle purchased with the industry average for a similar product. | Are products designed to last longer? | - |
| Time and number of products needed for disassembly. | Are the products designed to be disassembled? | - |
| Proportion of recycled materials in new products. | Are recycled materials included in the product design? | - |
| Proportion of materials that offer a safe recycling possibility. | Do materials designed to be recycled prevent pollution from recycling cycles? | - |
| Use of materials for production compared to GDP (potentially by sector). | Is Europe using fewer materials for production? | + |
| Input of substances that are classified as hazardous. | Does Europe use less volume and a lower number of environmentally hazardous substances in production? | + |
| Generation of waste (in production activities) (Indicated in EEA CSI01/WST004). | Does Europe generate less waste in production processes? | ++ |
| Participation of companies in networks of circular companies. | Are companies' strategies being adapted towards circular concepts such as remanufacturing and service-based offerings? | - |
| Environmental footprint of consumption (including materials) in Europe. | Are European citizens changing their consumption patterns towards greener goods and services? | + |
| Average of the real durability of selected products. | Do European citizens use the products for longer? | - |
| Waste generation (in consumer activities) (EEA indicator CSI041/WST0D4). | Does European consumption generate less waste? | ++ |
| Recycling rate for different types of waste/materials (EEA indicator WST005). | Is increasingly more waste being recycled? | ++ |
| Recycling material quality compared to virgin material quality. | To what extent do materials retain their value in recycling processes, avoiding under-recycling? | - |
| Environmental effects and cost-benefit analysis of municipal waste management in Europe. | To what extent is the recycling system optimized to achieve environmental and economic sustainability? | + |

(+ = higher data availability; - = lower data availability).

## 6. Conclusions

This paper has analyzed the proposals made by various researchers on indicators to measure the efficiency of the application of CE principles. This study has highlighted a number of issues:

(i) problems in establishing indicators, (ii) difficulty in defining the indicator, and (iii) the impossibility of obtaining the data. These issues open up possible new avenues of research.

Various researchers and businessmen encounter difficulties such as the selection and measurement of efficiency indicators when measuring levels of efficiency in CE [52].

This study points out a possible way to solve the above-mentioned issues. The most representative indicators measuring CE efficiency, as well as the availability of data to measure such efficiency, have been considered, using the availability data presented by COTEC [81].

The information contained in the indicators defined in this work is a relevant tool for managers to measure the efficiency of CE in tasks inherent to the planning process, such as the monitoring and control of actions implemented, decision-making, and comparative analysis in time and space. This paper reflects, through a concise set of indicators, the main elements of CE, which will facilitate the measurement of efficiency models in CE.

At the same time, a series of solutions is offered through the pertinent bibliographical review, for those problems that have arisen at the time of measuring the levels of efficiency in CE such as, for example, how a model of efficiency in CE should be considered or which dimensions are the most relevant when making a study on the subject of the present work. Therefore, this work is of special relevance because it provides information for external users and researchers regarding a series of indicators that allow measuring efficiency in CE, as well as the availability of these indicators regarding obtaining them.

**Author Contributions:** Conceptualization, J.S.-O., R.D.R.-S. and T.G.-V.; Formal analysis, V.R.-C., R.D.R.-S. and T.G.-V.; Investigation, R.D.R.-S., J.S.-O. and T.G.-V.; Methodology, J.S.-O.; Supervision, V.R.-C., R.D.R.-S. and T.G.-V.; Writing—original draft, J.S.-O. All authors have read and agreed to the published version of the manuscript.

**Funding:** This research was funded by Proyectos FEDER, grant number sol-201800103353-tra" and "The APC was funded by Junta de Andalucía (Proyectos FEDER)".

**Acknowledgments:** This paper has been funded by Proyecto de Excelencia Junta de Andalucía (FEDER). Economía Circular y Eficiencia. Hacia Nuevos modelos económicos (sol-201800103353-tra).

**Conflicts of Interest:** The authors declare no conflict of interest.

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
