# Peer review of "Indicators to Measure Efficiency in Circular Economies"

_sustainability, doi:10.3390/su12114483_

Round 1

Reviewer 1 Report

The research is appropriate design and the methods are adequately described.
The results are clearly presented and the conclusions are well supported by the results.

Author Response

We would like to express our gratitude for the excellent review that we have received.

Reviewer 2 Report

I find the paper very interesting and needed. It is always good to have been able to serach for papers that include the literature review on very up to date topics such as Circular Economy.

Author Response

(The authors gave the same response as above.)

Reviewer 3 Report

The text has sufficient quality to be published although it would improve if they showed the method followed to perform the submitted bibliographic search.

Author Response

I have included the following sentences about methods:

"For this, we have searched for papers cited in JCR (Journal Citations Reports) and Scopus using the keywords "circular economy, DEA". In addition, we have selected the papers with the highest number of citations".